# EXPANDNETS: LINEAR OVER-PARAMETERIZATION TO TRAIN COMPACT CONVOLUTIONAL NETWORKS

## ABSTRACT

In this paper, we introduce a novel approach to training a given compact network. To this end, we build upon over-parameterization, which typically improves both optimization and generalization in neural network training, while being unnecessary at inference time. We propose to expand each linear layer of the compact network into multiple linear layers, without adding any nonlinearity. As such, the resulting expanded network can benefit from over-parameterization during training but can be compressed back to the compact one algebraically at inference. As evidenced by our experiments, this consistently outperforms training the compact network from scratch and knowledge distillation using a teacher. In this context, we introduce several expansion strategies, together with an initialization scheme, and demonstrate the benefits of our ExpandNets on several tasks, including image classification, object detection, and semantic segmentation.

## 1 INTRODUCTION

With the growing availability of large-scale datasets and advanced computational resources, deep learning has achieved tremendous success in a variety of computer vision tasks, such as image classification (Krizhevsky et al., 2012; He et al., 2016), object detection (Ren et al., 2015; Redmon & Farhadi, 2018; 2016) and semantic segmentation (Long et al., 2015; Ronneberger et al., 2015). Over the past few years, "Wider and deeper are better" has become the rule of thumb to design network architectures (Simonyan & Zisserman, 2015; Szegedy et al., 2015; He et al., 2016; Huang et al., 2017). This trend, however, raises memory- and computation-related challenges, especially in the context of constrained environments, such as embedded platforms.

Deep and wide networks are well-known to be heavily over-parameterized, and thus a compact network, both shallow and thin, should often be sufficient. Unfortunately, compact networks are notoriously hard to train from scratch. As a consequence, designing strategies to train a given compact network has become increasingly popular, the most popular approach consisting of transferring the knowledge of a deep teacher network to the compact one of interest (Hinton et al., 2015; Romero et al., 2014; Yim et al., 2017a; Zagoruyko & Komodakis, 2017; Passalis & Tefas, 2018).

In this paper, we introduce an alternative approach to training compact neural networks, complementary to knowledge transfer. To this end, building upon the observation that network over-parameterization improves both optimization and generalization (Arora et al., 2018; Zhang et al., 2017; Reed, 1993; Allen-Zhu et al., 2018a;b; Kenji Kawaguchi & Bengio, 2018), we propose to increase the number of parameters of a given compact network by incorporating additional layers. However, instead of separating every two layers with a nonlinearity, as in most of the deep learning literature, we advocate introducing consecutive *linear* layers. In other words, we expand each linear layer of a compact network into a succession of multiple linear layers, without any nonlinearity in between. Since consecutive linear layers are equivalent to a single one (Saxe et al., 2014), such an expanded network, or ExpandNet, can be algebraically compressed back to the original compact one without any information loss.

While the use of successive linear layers appears in the literature, existing work (Baldi & Hornik, 1989; Saxe et al., 2014; Kawaguchi, 2016; Laurent & von Brecht, 2018; Zhou & Liang, 2018; Arora et al., 2018) has been mostly confined to networks without *any* nonlinearities and the theoretical study of their behavior under statistical assumptions that typically do not hold in practice. In particular, these studies have attempted to understand the learning dynamics and the loss landscapes of

deep networks. Here, by contrast, we focus on *practical, nonlinear, compact convolutional* neural networks, and investigate the use of linear expansion to introduce over-parameterization and improve training, so as to allow such networks to achieve better performance.

Specifically, as illustrated by Figure 1, we introduce three ways to expand a compact network: (i) replacing a fully-connected layer with multiple ones; (ii) replacing a $k \times k$ convolution by three convolutional layers with kernel size $1 \times 1$, $k \times k$ and $1 \times 1$, respectively; and (iii) replacing a $k \times k$ convolution with $k > 3$ by multiple $3 \times 3$ convolutions. Our experiments demonstrate that expanding convolutions, which has not been done before, is key to obtaining more effective compact networks.

Furthermore, we introduce a natural initialization strategy for our ExpandNets. Specifically, an ExpandNet has a nonlinear counterpart, with an identical number of parameters, obtained by adding nonlinearity between every two consecutive linear layers. We therefore propose to directly transfer the weights of this nonlinear model to the ExpandNet for initialization. We show that this yields a further performance boost for the resulting compact network.

In short, our contributions are (i) a novel approach to training given compact, nonlinear

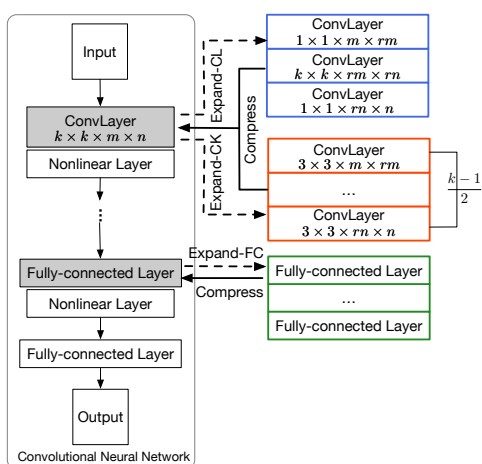

Figure 1: **ExpandNets.** We study 3 strategies to linearly expand a compact network. The resulting expanded network can be compressed back to the compact one algebraically, and consistently outperforms training the compact one from scratch and with knowledge distillation.

convolutional networks by expanding their linear layers; (ii) two strategies to expand convolutional layers; and (iii) an effective initialization scheme for the resulting ExpandNets. We demonstrate the benefits of our approach on several tasks, including image classification on ImageNet, object detection on PASCAL VOC and image segmentation on Cityscapes. Our ExpandNets consistently outperform training the corresponding compact networks from scratch and using knowledge distillation. We further analyze the benefits of linear over-parametrization on training via a series of experiments studying generalization, gradient confusion and the loss landscapes. We will make our code publicly available.

## 2 EXPANDNETS

Let us now introduce our approach to training compact networks by expanding their linear layers. In particular, we propose to make use of linear expansion, such that the resulting ExpandNet is equivalent to the original compact network and can be compressed back to the original structure algebraically. Below, we describe three different expansion strategies, starting with the simplest case of fully-connected layers, followed by two ways to expand convolutional layers.

**Expanding fully-connected layers.** The weights of a fully-connected layer can easily be represented in matrix form. Therefore, expanding such layers can be done in a straightforward manner by relying on matrix product. Specifically, let $\boldsymbol{W}_{n \times m}$ be the parameter matrix of a fully-connected layer with $m$ input channels and $n$ output ones. That is, given an input vector $\boldsymbol{x} \in \mathbb{R}^m$, the output $\boldsymbol{y} \in \mathbb{R}^n$ can be obtained as $\boldsymbol{y} = \boldsymbol{W}_{n \times m}\boldsymbol{x}$. Note that we ignore the bias, which can be taken into account by incorporating an additional channel with value 1 to $\boldsymbol{x}$. Expanding such a fully-connected layer with an arbitrary number of linear layers can be achieved by observing that its parameter matrix can be equivalently written as

$$\boldsymbol{W}_{n \times m} = \boldsymbol{W}_{n \times p_l} \times \boldsymbol{W}_{p_l \times p_{l-1}} \times \cdots \times \boldsymbol{W}_{p_1 \times m} \ . \tag{1}$$

While this yields an over-parametrization, it does not affect the underlying model capacity. More importantly, this allows us to increase not only the number of layers, but also the number of channels by setting $p_i > n, m$ for all $i$. To this end, we rely on the notion of *expansion rate*. Specifically, for an expansion rate $r$, we define $p_1 = rm$ and $p_i = rn, \forall i \neq 1$. Note that other strategies

are possible, e.g., $p_i = r^i m$, but ours has the advantage of preventing the number of parameters from exploding. In practice, considering the computational complexity of fully-connected layers, we advocate expanding each layer into only two or three layers with a small expansion rate.

**Expanding convolutional layers.** The operation performed by a convolutional layer can also be expressed in matrix form, by vectorizing the input tensor and defining a highly structured matrix whose elements are obtained from the vectorized convolutional filters. While this representation could therefore allow us to use the same strategy as with fully-connected layers, using arbitrary intermediate matrices would ignore the convolution structure, and thus alter the original operation performed by the layer. For a similar reason, one cannot simply expand a convolutional layer with kernel size $k \times k$ with a series of $k \times k$ convolutions, since, unless $k = 1$, the resulting receptive field size would differ from the original one.

To overcome this, we note that $1 \times 1$ convolutions retain the computational benefits of convolutional layers while not modifying the receptive field size. As illustrated in Figure 1, we therefore propose to expand a $k \times k$ convolutional layer into 3 consecutive convolutional layers: a $1 \times 1$ convolution; a $k \times k$ one; and another $1 \times 1$ one. This strategy still allows us to increase the number of channels in the intermediate layer. Specifically, for an original layer with $m$ input channels and $n$ output ones, given an expansion rate $r$, we define the number of output channels of the first $1 \times 1$ layer as $p = rm$ and the number of output channels of the intermediate $k \times k$ layer as $q = rn$.

Compressing an expanded convolutional layer into the original one can still be done algebraically. To this end, one can reason with the matrix representation of convolutions. For an input tensor of size $(w \times h \times m)$, the matrix representation of the original layer can be recovered as

$$\boldsymbol{W}_{nv \times mv} = \boldsymbol{W}_{nv \times qv} \times \boldsymbol{W}_{qv \times pv} \times \boldsymbol{W}_{pv \times mv} \ , \tag{2}$$

where $v = w \cdot h$ and each intermediate matrix has a structure encoding a convolution. It can be verified that the resulting matrix will also have a convolution structure, with filter size $k \times k$.

**Expanding convolutional kernels.** While $3 \times 3$ kernels have become increasingly popular in very deep architectures (He et al., 2016), larger kernel sizes are often exploited in compact networks, so as to increase their expressiveness and their receptive fields. As illustrated in Figure 1, $k \times k$ kernels with $k > 3$ can be equivalently represented with a series of $l$ $3 \times 3$ convolutions, where $l = \frac{k-1}{2}$. As before, the number of channels in the intermediate $3 \times 3$ layers can be larger than that in the original $k \times k$ one, thus allowing us to linearly over-parametrize the model. Similarly to the fully-connected case, for an expansion rate $r$, we set the number of output channels of the first $3 \times 3$ layer to $p_1 = rm$ and that of the subsequent layers to $p_i = rn$. The same matrix-based strategy as before can then be used to algebraically compress back the expanded kernels.

**Dealing with convolution padding and strides.** In modern convolutional networks, padding and strides are widely used to retain information from the input feature map while controlling the size of the output one. To expand a convolutional layer with padding $p$, we propose to use padding $p$ in the first layer of the expanded unit while not padding the remaining layers. Furthermore, to handle a stride $s$, when expanding convolutional layers, we set the stride of the middle layer to $s$ and of the others to 1. When expanding convolutional kernels, we use a stride 1 for all layers except for the last one whose stride is set to $s$. These two strategies guarantee that the resulting ExpandNet can be compressed back to the original compact model without any information loss.

Overall, the strategies discussed above allow us to expand an arbitrary compact network into an equivalent deeper and wider one. Note that these strategies can be used independently or together. In any event, once the resulting ExpandNet is trained, it can be compressed back to the original compact architecture in a purely algebraic manner, that is, at absolutely no loss of information.

## 2.1 INITIALIZING EXPANDNETS

As will be demonstrated by our experiments, training an ExpandNet from scratch yields consistently better results than training the original compact network. However, in the context of deep networks, initialization can have an important effect on the final results (Mishkin & Matas, 2015; He et al., 2015). While designing an initialization strategy specifically for compact networks is an unexplored and challenging research direction, our ExpandNets can be initialized in a very natural manner.

To this end, we exploit the fact that an ExpandNet has a natural nonlinear counterpart, which can be obtained by incorporating a nonlinear activation function between each pair of linear layers. We therefore propose to initialize the parameters of an ExpandNet by simply training its nonlinear counterpart and transferring the resulting parameters to the ExpandNet. The initialized ExpandNet is then trained in the standard manner. As evidenced by our experiments below, when the nonlinear counterpart achieves better performance than the compact network, this simple strategy typically yields an additional accuracy boost to our approach.

## 3 EXPERIMENTS

In this section, we demonstrate the benefits of our ExpandNets on several tasks, including image classification, object detection, and semantic segmentation. We then analyze the training behavior of our different expansion strategies.

We denote the expansion of fully-connected layers by *FC*, of convolutional layers by *CL*, and of convolutional kernels by *CK*. When combining convolutional expansions with fully-connected ones, we use *CL+FC* or *CK+FC*, and add *+Init* to indicate the use of our initialization strategy.

### 3.1 IMAGE CLASSIFICATION

We first study the use of our approach with very small networks on CIFAR-10 and CIFAR-100, and then turn to the more challenging ImageNet dataset, where we show that our method can improve the results of the compact MobileNet (Howard et al., 2017), MobileNetV2 (Sandler et al., 2018) and ShuffleNetV2 $0.5\times$ (Ma et al., 2018) architectures.

#### 3.1.1 CIFAR-10 AND CIFAR-100

**Experimental setup.** For CIFAR-10 and CIFAR-100, we use the same compact network as in (Passalis & Tefas, 2018). It is composed of 3 convolutional layers with $3 \times 3$ kernels and no padding. These 3 layers have 8, 16 and 32 output channels, respectively. Each of them is followed by a batch normalization layer, a ReLU layer and a $2 \times 2$ max pooling layer. The output of the last layer is passed through a fully-connected layer with 64 units, followed by a logit layer with either 10 or 100 units. To evaluate our kernel expansion method, we also report results obtained with a similar network where the $3 \times 3$ kernels were replaced by $7 \times 7$ ones, with a padding of 3. All networks, including our ExpandNets, were trained for 150 epochs using a batch size of 128. We used standard stochastic gradient descent (SGD) with a momentum of 0.9 and a learning rate of 0.01, divided by 10 at epochs 50 and 100. With this strategy, all networks reached convergence. In this set of experiments, the expand rate $r$ is always set to 4 to balance the accuracy-efficiency trade-off. We evaluate the influence of this parameter in Appendix A.3. Note that our ExpandNet strategy is complementary to knowledge transfer; that is, we can apply any typical knowledge transfer method using our ExpandNet as student instead of the compact network directly. To demonstrate the benefits of ExpandNets in this scenario, we conduct experiments using knowledge distillation (KD) (Hinton et al., 2015), hint-based transfer (Hint)(Romero et al., 2014) or probabilistic knowledge transfer (PKT) (Passalis & Tefas, 2018) using a ResNet18 teacher network.

**Results.** We focus here on the SmallNet with $7 \times 7$ kernels, for which we can evaluate all our expansion strategies, including the CK ones, and report the results obtained with the model with $3\times3$ kernels in Table 6 in Appendix A.1. The results of our all networks with and without knowledge distillation[1] are provided in Table 1. As shown in the top portion of the table, only expanding the fully-connected layer yields mild improvement. However, expanding the convolutional ones clearly outperforms over the compact network, and is further boosted by expanding the fully-connected one and by using our initialization scheme. Note that expanding the kernels yields higher accuracy, with ExpandNet-CK+FC+Init achieving the best results. Note that even without KD, our ExpandNets outperform the SmallNet with KD. The gap is further increased when we also use KD, as can be seen in the bottom portion of the table.

---

[1]Having found KD to be the most effective knowledge transfer strategy,

Table 1: Top-1 accuracy (%) of SmallNet with $7 \times 7$ kernels vs ExpandNets with $r = 4$ on CIFAR-10 and CIFAR-100.

| Network | Transfer | CIFAR-10 | CIFAR-100 |
|---|---|---|---|
| SmallNet | *w/o* KD | $78.63 \pm 0.41$ | $46.63 \pm 0.27$ |
| SmallNet | *w/* KD | $78.97 \pm 0.37$ | $47.04 \pm 0.35$ |
| ExpandNet-FC | | $78.64 \pm 0.39$ | $46.59 \pm 0.45$ |
| ExpandNet-CL | | $78.47 \pm 0.20$ | $46.90 \pm 0.66$ |
| ExpandNet-CL+FC | | $79.11 \pm 0.23$ | $46.66 \pm 0.43$ |
| ExpandNet-CL+FC+Init | *w/o* KD | $79.98 \pm 0.28$ | $47.98 \pm 0.48$ |
| ExpandNet-CK | | $80.27 \pm 0.24$ | $48.55 \pm 0.51$ |
| ExpandNet-CK+FC | | $80.31 \pm 0.27$ | $48.62 \pm 0.47$ |
| ExpandNet-CK+FC+Init | | $\mathbf{80.81 \pm 0.27}$ | $\mathbf{49.82 \pm 0.25}$ |
| ExpandNet-CL+FC | | $79.60 \pm 0.25$ | $47.41 \pm 0.51$ |
| ExpandNet-CL+FC+Init | *w/* KD | $80.29 \pm 0.25$ | $48.62 \pm 0.34$ |
| ExpandNet-CK+FC | | $80.63 \pm 0.31$ | $49.13 \pm 0.45$ |
| ExpandNet-CK+FC+Init | | $\mathbf{81.21 \pm 0.17}$ | $\mathbf{50.37 \pm 0.39}$ |

Table 2: Top-1 accuracy (%) on the ILSVRC2012 validation set (ExpandNets with $r = 4$).

| Model | ImageNet (90 epochs) | ImageNet (400 epochs) |
|---|---|---|
| MobileNet | 66.48 | 70.47 |
| MobileNet (*w/* KD) | 69.01 | *N/A* |
| ExpandNet-CL | 69.40 | **70.82** |
| ExpandNet-CL(*w/* KD) | **70.47** | *N/A* |
| MobileNetV2 | 63.75 | 70.67 |
| MobileNetV2 (*w/* KD) | 65.40 | *N/A* |
| ExpandNet-CL | 65.62 | **70.70** |
| ExpandNet-CL (*w/* KD) | **67.19** | *N/A* |
| ShuffleNetV2 0.5× | | 56.89 |
| ShuffleNetV2 0.5× (*w/* KD) | | 57.59 |
| ExpandNet-CL | | 57.38 |
| ExpandNet-CL (*w/* KD) | | **57.68** |

### 3.1.2 IMAGENET

We now turn to the more challenging case of ImageNet (Russakovsky et al., 2015). In this case, for MobileNets, we perform data augmentation by taking random image crops and resizing them to $224 \times 224$ pixels. We further perform random horizontal flips and subtract the per-pixel mean.

**Experimental setup.** For this set of experiments, we make use of the compact MobileNet (Howard et al., 2017), MobileNetV2 (Sandler et al., 2018) and ShuffleNetV2 (Ma et al., 2018) models, which were designed to be compact and yet achieve good results. We rely on a pytorch implementation of these models. For our approach, we use our CL strategy to expand all convolutional layers with kernel size of $3 \times 3$ in MobileNet and ShuffleNet, while only expanding the non-residual $3 \times 3$ convolutional layers in MobileNetV2. We trained the MobileNets in 2 regimes: Short-term (as for ResNets (He et al., 2016)), where we trained for 90 epochs with a weight decay of 0.0001 and an initial learning rate of 0.1, divided by 10 every 30 epochs; Long-term, where we trained for 400 epochs with a weight decay of 0.00004 and an initial learning rate of 0.01, divided by 10 at epochs 200 and 300. In all cases, we employed SGD with a momentum of 0.9 and a batch size of 256. For ShuffleNet, we use of a small ShuffleNetV2 $0.5 \times$, trained in the same manner as in (Ma et al., 2018). We also perform KD from a ResNet152 (78.32% top-1 accuracy), tuning the KD hyper-parameters to the best accuracy for each method.

**Results.** We compare the results of the original models with those of our expanded versions in Table 2. Our expansion strategy yields a huge improvement, especially in the short-term regime, increasing the top-1 accuracy of MobileNet, MobileNetV2 and ShuffleNet by 2.92, 1.87 and 0.49 percentage points (pp). This short-term regime could be valuable to work with limited access to resources. In the long-term regime, our expansion strategy nonetheless still achieves an improvement. Furthermore, our vanilla ExpandNets outperform the compact models with KD, even though we do not require a teacher. When also using KD, ExpandNets can achieve even better performance. Note that, on this dataset, their nonlinear counterparts do not outperform original models.

## 3.2 OBJECT DETECTION

Our approach is not restricted to image classification. We first demonstrate its benefits in the context of one-stage object detection using the PASCAL VOC dataset (Everingham et al., a;b). To this end, we trained networks on the PASCAL VOC2007 + 2012 training and validation sets and report the mean average precision (mAP) on the PASCAL VOC2007 test set.

**Experimental setup.** We make use of YOLO-LITE (Huang et al., 2018), which was designed to work in constrained environments. It consists of a backbone part and a head part, and we only expand the convolutional layers in the backbone. YOLO-LITE is a very compact network, with only 5 convolutional layers in the

Table 3: mAPs (%) of YOLO-LITE vs Expand-Nets on the PASCAL VOC2007 test set.

| Model | VOC2007 Test |
|---|---|
| YOLO-LITE | 27.34 |
| ExpandNet-CL | 30.97 |
| ExpandNet-CL+Init | **35.14** |

backbone, each followed by a batch normalization layer, a leaky-ReLU layer and a $2 \times 2$ max pooling layer. We expand all 5 convolutional layers using our CL strategy with $r = 4$. We trained all networks in the standard YOLO fashion (Redmon & Farhadi, 2018; 2016).

**Results.** The results on the PASCAL VOC 2007 test set are reported in Table 3. They show that, as for image recognition, our expansion strategy yields better performance. Since YOLO-LITE is very compact, our initialization scheme boosts the performance by over 7pp.

## 3.3 SEMANTIC SEGMENTATION

We then demonstrate the benefits of our approach on the task of semantic segmentation using the Cityscapes dataset (Cordts et al., 2016). Cityscapes is a large-scale dataset containing 5000 images of $1024 \times 2048$. Following the standard protocol, we report the mean Intersection over Union (mIoU), mean recall (mRec) and mean precision (mPrec).

**Experimental setup.** For this experiment, we rely on the U-Net (Ronneberger et al., 2015), which is a relatively compact network consisting of a contracting path and an expansive path. The contracting path follows the typical architecture of a convolutional network. It consists of blocks of two $3 \times 3$ convolutions, each followed by a ReLU, and a $2 \times 2$ max pooling. The number of channels in each block is 32, 64, 128, 256, 512, respectively. We apply our CL expansion strategy with $r = 4$ to all convolutions in the contracting path. We train the networks on 4 GPUs using the standard SGD optimizer with a momentum of 0.9 and a learning rate of $1e - 8$.

Table 4: U-Net vs ExpandNets with $r = 4$ on the Cityscapes validation set.

| Model | mIoU | mRec | mPrec |
|---|---|---|---|
| U-Net | 56.59 | 74.29 | 65.11 |
| ExpandNet-CL | **57.85** | **76.53** | **65.94** |

**Results.** Our results on the Cityscapes validation set are shown in Table 4. Note that our ExpandNet also outperforms the original compact U-Net on this task.

## 3.4 ANALYSIS OF OUR APPROACH

To further analyze our approach, we study its behavior during training and its generalization ability. For these experiments, we make use of the CIFAR-10 and CIFAR-100 datasets, and use the same settings as in Appendix A.3. In Appendix A.4, we also showcase the use of our approach with the larger AlexNet architecture on ImageNet. Finally, we evaluate the complexity of the models in terms of number of parameters, FLOPs, and training and testing inference speed in Appendix B. Since our ExpandNets can be compressed back to the original networks, *at test time, they have exactly the same number of parameters, FLOPS, and inference time, but achieve higher accuracy*.

### 3.4.1 TRAINING BEHAVIOR

To investigate the benefits of linear over-parameterization on training, we make use of the gradient confusion introduced by (Sankararaman et al., 2019) to show that the gradients of nonlinearly over-parameterized networks were more consistent across mini-batches. Specifically, following (Sankararaman et al., 2019), we measure gradient confusion (or rather consistency) as the minimum cosine similarity from 100 randomly-sampled pairs of mini-batches at the end of each training epoch. As in (Sankararaman et al., 2019), we also combine the gradient cosine similarity of 100 pairs of sampled mini-batches at the end of training from each independent run and perform Gaussian kernel density estimation on this data. While, during training, we aim for the minimum cosine similarity to be high, at the end of training, the resulting distribution should be peaked around zero.

We run each experiment 5 times and show the average values across all runs in Figure 2. From the training and test curves, we observe that our ExpandNets-CL/CK speed up convergence and yield a smaller generalization error gap. They also yield lower gradient confusion (higher minimum cosine similarity) and a more peaked density of pairwise gradient cosine similarity. This indicates that our ExpandNets-CL/CK are easier to train than the compact model. By contrast, only expanding the fully-connected layers does not facilitate training. Additional plots are provided in Appendix C.

### 3.4.2 GENERALIZATION ABILITY

We then analyse the generalization ability of our approach. To this end, we first study the loss landscapes using the method in (Li et al., 2018). As shown in Figure 3, our ExpandNets, particularly with convolutional expansion, produce flatter minima, which, as discussed in (Li et al., 2018), indicates better generalization.

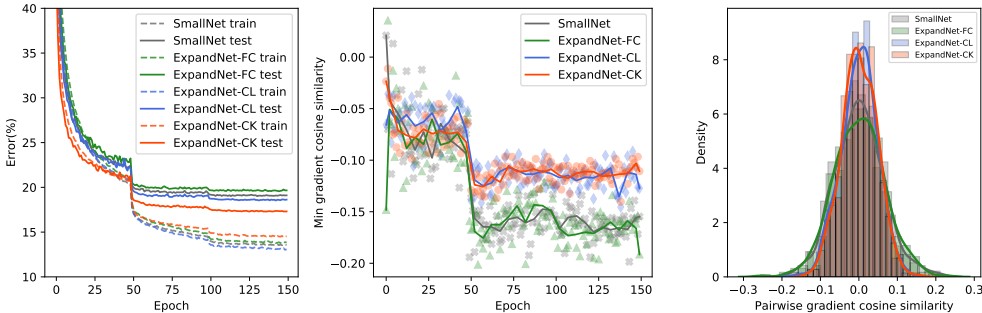

Figure 2: **Training behavior of networks with** $7 \times 7$ **kernels on CIFAR-10.** *Left:* Training and test curves over 150 epochs. *Middle:* Minimum pairwise gradient cosine similarity at the end of each training epoch. *Right:* Kernel density estimation of pairwise gradient cosine similarity at the end of training (over 5 independent runs).

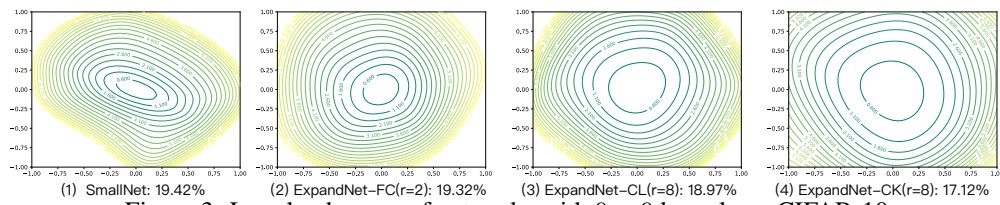

(1) SmallNet: 19.42%    (2) ExpandNet–FC(r=2): 19.32%    (3) ExpandNet–CL(r=8): 18.97%    (4) ExpandNet–CK(r=8): 17.12%

Figure 3: Loss landscapes of networks with $9 \times 9$ kernels on CIFAR-10.

Table 5: **Generalization ability on Corrupted CIFAR-10.** We report the top-1 error (%). Note that our ExpandNets yield larger generalization gaps than the compact network in almost all cases involving convolutional expansion. By contrast expanding FC layers often does not help.

| Dataset | Model | Kernel size $k$ | | | | | |
|---|---|---|---|---|---|---|---|
| | | 5 | | | 9 | | |
| | | Best Test | Last Test | Train | Best Test | Last Test | Train |
| 20% | SmallNet | $20.90 \pm 0.16$ | $21.09 \pm 0.20$ | $32.05 \pm 0.31$ | $22.56 \pm 0.39$ | $22.93 \pm 0.18$ | $29.61 \pm 0.36$ |
| | ExpandNet-FC | $20.87 \pm 0.29$ | $21.06 \pm 0.26$ | $32.04 \pm 0.12$ | $22.95 \pm 0.39$ | $23.48 \pm 0.38$ | $29.83 \pm 0.34$ |
| | ExpandNet-CL | $20.47 \pm 0.46$ | $20.62 \pm 0.43$ | $31.80 \pm 0.23$ | $22.13 \pm 0.49$ | $22.73 \pm 0.53$ | $29.76 \pm 0.19$ |
| | ExpandNet-CK | $\mathbf{19.42 \pm 0.20}$ | $\mathbf{19.63 \pm 0.17}$ | $31.55 \pm 0.25$ | $\mathbf{19.32 \pm 0.31}$ | $\mathbf{19.55 \pm 0.30}$ | $\mathbf{31.65 \pm 0.17}$ |
| 50% | SmallNet | $25.38 \pm 0.45$ | $25.68 \pm 0.52$ | $54.49 \pm 0.41$ | $28.64 \pm 0.46$ | $30.44 \pm 0.57$ | $52.67 \pm 0.45$ |
| | ExpandNet-FC | $25.36 \pm 0.63$ | $25.71 \pm 0.77$ | $54.44 \pm 0.08$ | $28.46 \pm 0.43$ | $30.89 \pm 0.38$ | $52.51 \pm 0.36$ |
| | ExpandNet-CL | $24.27 \pm 0.33$ | $24.63 \pm 0.44$ | $54.29 \pm 0.24$ | $27.42 \pm 0.35$ | $29.28 \pm 0.50$ | $52.67 \pm 0.27$ |
| | ExpandNet-CK | $\mathbf{22.82 \pm 0.27}$ | $\mathbf{23.00 \pm 0.29}$ | $53.93 \pm 0.23$ | $\mathbf{22.77 \pm 0.14}$ | $\mathbf{22.99 \pm 0.15}$ | $\mathbf{54.37 \pm 0.12}$ |
| 80% | SmallNet | $37.99 \pm 0.64$ | $39.33 \pm 0.75$ | $76.14 \pm 0.15$ | $41.73 \pm 0.58$ | $47.96 \pm 1.07$ | $74.01 \pm 0.32$ |
| | ExpandNet-FC | $38.35 \pm 0.59$ | $39.61 \pm 0.87$ | $76.51 \pm 0.15$ | $42.31 \pm 0.46$ | $49.36 \pm 1.44$ | $74.59 \pm 0.35$ |
| | ExpandNet-CL | $36.75 \pm 0.64$ | $38.08 \pm 0.50$ | $76.09 \pm 0.11$ | $41.44 \pm 0.46$ | $46.75 \pm 0.49$ | $74.46 \pm 0.08$ |
| | ExpandNet-CK | $\mathbf{33.29 \pm 1.04}$ | $\mathbf{34.24 \pm 0.85}$ | $75.77 \pm 0.22$ | $\mathbf{33.29 \pm 0.58}$ | $\mathbf{33.75 \pm 0.49}$ | $\mathbf{76.27 \pm 0.23}$ |

As a second study of generalization, we evaluate the memorization ability of our ExpandNets on corrupted datasets, as suggested in (Zhang et al., 2017). To this end, we utilize the open-source implementation of (Zhang et al., 2017) to generate three CIFAR-10 and CIFAR-100 training sets, containing 20%, 50% and 80% of random labels, respectively, while the test set remains clean.

In Table 5 (and Tables 10, 11 and 12 in Appendix A.5), we report the top-1 test errors of the best and last models, as well as the training errors of the last model. These results evidence that expanding convolutional layers and kernels typically yields lower test errors and higher training ones, which implies that our better results in the other experiments are not due to simply memorizing the datasets, but truly to better generalization ability.

## 4 RELATED WORK

Very deep neural networks currently constitute the state of the art for many computer vision tasks. These networks, however, are known to be heavily over-parameterized, and making them smaller would facilitate their use in resource-constrained environments, such as embedded platforms. As a consequence, much research has recently been devoted to developing more compact architectures.

*Network compression* constitutes one of the most popular trends in this area. In essence, it aims to reduce the size of a pre-trained very deep network while losing as little accuracy as possible, or even none at all. In this context, existing approaches can be roughly grouped into two categories: (i) parameter pruning and sharing (LeCun et al., 1990; Han et al., 2015; Courbariaux et al., 2016;

M Figurnov, 2016; Pavlo Molchanov, 2017; Ullrich et al., 2017; Carreira-Perpin & Idelbayev, 2018), which aims to remove the least informative parameters; and (ii) low-rank factorization (Denil et al., 2013; Sainath et al., 2013; Lebedev et al., 2014; Jin et al., 2015; Liu et al., 2015), which uses decomposition techniques to reduce the size of the parameter matrix/tensor in each layer. While compression is typically performed as a post-processing step, it has been shown that incorporating it during training could be beneficial (Alvarez & Salzmann, 2016; 2017; Wen et al., 2016; 2017). In any event, even though compression reduces a network's size, it does not provide one with the flexibility of designing a network with a specific architecture and of a specific size. Furthermore, it often produces networks that are much larger than the ones we consider here, e.g., compressed networks with $O(1M)$ parameters vs $O(10K)$ for the SmallNets used in our experiments.

In a parallel line of research, several works have proposed design strategies to reduce a network's number of parameters (Wu et al., 2016; Szegedy et al., 2016; Howard et al., 2017; Romera et al., 2018; Sandler et al., 2018). Again, while more compact networks can indeed be developed with these mechanisms, they impose constraints on the network architecture, and thus do not allow one to simply train a given compact network. Furthermore, as shown by our experiments, our approach is complementary to these works. For example, we can improve the results of MobileNetV2 (Sandler et al., 2018) by training it using our expansion strategy.

Here, in contrast to the above-mentioned literature on compact networks, we seek to train a *given* compact network with an arbitrary architecture. This is also the task addressed by *knowledge transfer* approaches. To achieve this, existing methods leverage the availability of a pre-trained very deep teacher network. In (Hinton et al., 2015), this is done by defining soft labels from the logits of the teacher; in (Romero et al., 2014), (Zagoruyko & Komodakis, 2017) and (Yim et al., 2017b) by transferring intermediate representations, attention maps and Gram matrices, respectively, from the teacher to the compact network; in (Passalis & Tefas, 2018) by aligning the feature distributions of the deep and compact networks.

In this paper, we introduce an alternative strategy to train compact networks, complementary to knowledge transfer. Inspired by the theory showing that over-parameterization helps training (Arora et al., 2018; Zhang et al., 2017; Reed, 1993; Allen-Zhu et al., 2018a;b), we propose to expand each linear layer in a given compact network into a succession of multiple linear layers. Our experiments evidence that training such expanded networks, which can then be compressed back algebraically, yields better results than training the original compact networks, thus empirically confirming the benefits of over-parameterization. Our results also show that our approach outperforms knowledge transfer, even when not using a teacher network.

Note that linearly over-parametrized neural networks have been investigated both in the early neural network days (Baldi & Hornik, 1989) and more recently (Saxe et al., 2014; Kawaguchi, 2016; Laurent & von Brecht, 2018; Zhou & Liang, 2018; Arora et al., 2018). These methods, however, typically study purely linear networks, with a focus on the convergence behavior of training in this linear regime. For example, (Arora et al., 2018) showed that linear over-parametrization modifies the gradient updates in a unique way that speeds up convergence. Here, we propose to exploit the benefits of linear over-parametrization to improve the training of compact networks. To this end, in contrast to the above-mentioned methods which all focus on fully-connected layers, we develop two strategies to expand *convolutional* layers, as well as an approach to initializing our expanded networks, and empirically demonstrate their critical impact on prediction accuracy.

## 5 CONCLUSION

We have introduced an approach to training a given compact network from scratch by exploiting over-parametrization. Specifically, we have shown that over-parametrizing the network *linearly* facilitates the training of compact networks. Since the resulting networks are equivalent to the original compact ones, they can be compressed back at no loss of information. Our experiments have demonstrated that our ExpandNets consistently outperform the original compact networks. In particular, when applicable, our CK expansion strategy tends to yield the best results. When it doesn't, that is, for a kernel size of 3, the CL one nonetheless remains highly effective. Our technique is general and can also be used in conjunction with knowledge transfer approaches. Furthermore, our initialization scheme yields a consistent accuracy boost. This strategy, however, is not the only possible one. In the future, we will therefore aim to develop other initialization schemes for our ExpandNets, and generally for compact networks.

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

## A  COMPLEMENTARY EXPERIMENTS

We provide additional experimental results to further evidence the effectiveness of our three expansion strategies.

### A.1  SMALLNET WITH $3 \times 3$ KERNELS

As mentioned in the main paper, we also evaluate our approach using the same small network as in (Passalis & Tefas, 2018), with kernel size $3 \times 3$ instead of $7 \times 7$ in the main paper, on CIFAR-10 and CIFAR-100. In this case, we make use of our CL expansion strategy, with and without our initialization scheme, because the CK one does not apply to $3 \times 3$ kernels. We trained all these networks for 100 epochs using a batch size of 128. We used Adam with a learning rate of 0.001, divided by 10 at epoch 50, which matches the setup in (Passalis & Tefas, 2018). As reported in Table 6, expanding the convolutional layers yields higher accuracy than the small network. This is further improved by also expanding the fully-connected layer, and even more so when using our initialization strategy.

Table 6: **Top-1 accuracy** (%) **of SmallNet with** $3 \times 3$ **kernels vs ExpandNets with** $r = 4$ **on CIFAR-10 and CIFAR-100.** Our approach yields consistently better results than the compact network, particularly when expanding all layers and using our initialization strategy.

| Model | Transfer | CIFAR-10 | CIFAR-100 |
|---|---|---|---|
| SmallNet | *w/o KD* | $73.32 \pm 0.20$ | $40.40 \pm 0.60$ |
| SmallNet | *w/ KD* | $73.34 \pm 0.31$ | $40.46 \pm 0.56$ |
| ExpandNet-FC | | $73.78 \pm 0.83$ | $40.52 \pm 0.71$ |
| ExpandNet-CL | *w/o KD* | $73.96 \pm 0.30$ | $40.91 \pm 0.47$ |
| ExpandNet-CL+FC | | $74.45 \pm 0.29$ | $41.12 \pm 0.49$ |
| ExpandNet-CL+FC+Init | | $\mathbf{75.16 \pm 0.23}$ | $\mathbf{42.41 \pm 0.21}$ |
| ExpandNet-CL+FC | *w/ KD* | $74.52 \pm 0.37$ | $41.51 \pm 0.49$ |
| ExpandNet-CL+FC+Init | | $\mathbf{75.17 \pm 0.51}$ | $42.67 \pm 0.67$ |

### A.2  KNOWLEDGE TRANSFER WITH EXPANDNETS

In the main paper, we claim that our ExpandNet strategy is complementary to knowledge transfer. Following (Passalis & Tefas, 2018), on CIFAR-10, we make use of the ResNet18 as teacher. Furthermore, we also use the same compact network with kernel size $3 \times 3$ and training setting as in (Passalis & Tefas, 2018). In Table 7, we compare the results of different knowledge transfer strategies, including knowledge distillation (KD) (Hinton et al., 2015), hint-based transfer (Hint)(Romero et al., 2014) and probabilistic knowledge transfer (PKT) (Passalis & Tefas, 2018), applied to the compact network and to our ExpandNets, without and with our initialization scheme. Note that using our ExpandNets, with and without initialization, consistently outperforms using the compact network. Altogether, we therefore believe that, to train a given compact network, one should really use both knowledge transfer and our ExpandNets to obtain the best results.

Table 7: **Knowledge transfer from the ResNet18 on CIFAR-10.** Using ExpandNets as student networks yields consistently better results than directly using SmallNet.

| Network | Transfer | Top-1 Accuracy |
|---|---|---|
| SmallNet | Baseline | $73.32 \pm 0.20$ |
| SmallNet | KD | $73.34 \pm 0.31$ |
| | Hint | $33.71 \pm 4.35$ |
| | PKT | $68.36 \pm 0.35$ |
| ExpandNet | KD | $74.52 \pm 0.37$ |
| (CL+FC) | Hint | $52.46 \pm 2.43$ |
| | PKT | $70.97 \pm 0.70$ |
| ExpandNet | KD | $\mathbf{75.17 \pm 0.51}$ |
| (CL+FC+Init) | Hint | $58.27 \pm 3.83$ |
| | PKT | $71.65 \pm 0.41$ |

### A.3  ABLATION STUDY

In this section, we conduct an ablation study on the hyper-parameter choice, specifically the expansion rate and kernel size. We evaluate the behavior of our different expansion strategies, FC, CL and CK, separately, when varying the expansion rate $r \in \{2, 4, 8\}$ and the kernel size $k \in \{3, 5, 7, 9\}$. Compared to our previous CIFAR-10 and CIFAR-100 experiments, we use a deeper network with an

Table 8: **Small networks vs ExpandNets on CIFAR-10 (Top) and CIFAR-100 (Bottom).** We report the top-1 accuracy for the original compact networks and for different versions of our approach. Note that our ExpandNets yield higher accuracy than the compact network in almost all cases involving expanding convolutions cases. By contrast expanding FC layers does often not help.

| Model | $r$ | Kernel size $k$ | | | |
|---|---|---|---|---|---|
| | | 3 | 5 | 7 | 9 |
| SmallNet | | $79.34 \pm 0.42$ | $81.25 \pm 0.14$ | $81.44 \pm 0.20$ | $80.08 \pm 0.48$ |
| | 2 | $79.13 \pm 0.47$ | $81.26 \pm 0.33$ | $80.98 \pm 0.25$ | $80.43 \pm 0.22$ |
| ExpandNet-FC | 4 | $78.92 \pm 0.36$ | $81.13 \pm 0.46$ | $80.85 \pm 0.24$ | $80.13 \pm 0.29$ |
| | 8 | $79.64 \pm 0.41$ | $81.21 \pm 0.18$ | $80.75 \pm 0.45$ | $80.16 \pm 0.16$ |
| | 2 | $79.46 \pm 0.21$ | $81.50 \pm 0.31$ | $81.30 \pm 0.30$ | $80.26 \pm 0.66$ |
| ExpandNet-CL | 4 | $\mathbf{79.90 \pm 0.21}$ | $81.60 \pm 0.15$ | $81.15 \pm 0.36$ | $80.62 \pm 0.32$ |
| | 8 | $79.78 \pm 0.20$ | $\mathbf{81.75 \pm 0.40}$ | $\mathbf{81.53 \pm 0.33}$ | $\mathbf{80.78 \pm 0.25}$ |
| | 2 | $N/A$ | $81.72 \pm 0.31$ | $82.19 \pm 0.24$ | $81.60 \pm 0.11$ |
| ExpandNet-CK | 4 | $N/A$ | $82.34 \pm 0.43$ | $82.34 \pm 0.22$ | $81.73 \pm 0.33$ |
| | 8 | $N/A$ | $\mathbf{82.37 \pm 0.25}$ | $\mathbf{82.84 \pm 0.28}$ | $\mathbf{82.53 \pm 0.30}$ |
| SmallNet | | $48.14 \pm 0.29$ | $50.44 \pm 0.07$ | $49.62 \pm 0.50$ | $48.70 \pm 0.38$ |
| | 2 | $47.21 \pm 0.46$ | $48.39 \pm 0.77$ | $47.88 \pm 0.41$ | $46.36 \pm 0.34$ |
| ExpandNet-FC | 4 | $47.44 \pm 0.66$ | $48.92 \pm 0.47$ | $48.43 \pm 0.56$ | $46.90 \pm 0.34$ |
| | 8 | $47.55 \pm 0.25$ | $49.44 \pm 0.65$ | $48.66 \pm 0.49$ | $47.15 \pm 0.28$ |
| | 2 | $47.68 \pm 0.85$ | $50.39 \pm 0.45$ | $49.78 \pm 0.33$ | $48.68 \pm 0.70$ |
| ExpandNet-CL | 4 | $48.25 \pm 0.13$ | $50.68 \pm 0.27$ | $49.81 \pm 0.31$ | $\mathbf{48.87 \pm 0.65}$ |
| | 8 | $\mathbf{48.93 \pm 0.13}$ | $\mathbf{50.95 \pm 0.42}$ | $49.95 \pm 0.37$ | $48.85 \pm 0.42$ |
| | 2 | $N/A$ | $51.18 \pm 0.44$ | $51.09 \pm 0.41$ | $50.40 \pm 0.35$ |
| ExpandNet-CK | 4 | $N/A$ | $\mathbf{52.13 \pm 0.36}$ | $51.82 \pm 0.67$ | $50.62 \pm 0.65$ |
| | 8 | $N/A$ | $52.05 \pm 0.59$ | $\mathbf{52.48 \pm 0.54}$ | $\mathbf{51.57 \pm 0.15}$ |

extra convolutional layer with 64 channels, followed by a batch normalization layer, a ReLU layer and a $2 \times 2$ max pooling layer. We use SGD with a momentum of 0.9 and a weight decay of 0.0005 for 150 epochs. The initial learning rate was 0.01, divided by 10 at epoch 50 and 100. Furthermore, we used zero-padding to keep the size of the input and output feature maps of each convolutional layer unchanged. We use the same networks in our approach analysis experiments.

The results of these experiments are provided in Table 8. We observe that our different strategies to expand convolutional layers outperform the compact network in almost all cases, while only expanding fully-connected layers doesn't work well. In particular, for kernel sizes $k > 3$, ExpandNet-CK yields consistently higher accuracy than the corresponding compact network, independently of the expansion rate. For $k = 3$, where ExpandNet-CK is not applicable, ExpandNet-CL comes as an effective alternative, also consistently outperforming the baseline.

## A.4 WORKING WITH LARGER NETWORKS

We also evaluate the use of our approach with a larger network. To this end, we make use of AlexNet (Krizhevsky et al., 2012) on ImageNet. AlexNet relies on kernels of size 11 and 5 in its first two convolutional layers, which makes our CK expansion strategy applicable.

We use a modified, more compact version of AlexNet, where we replace the first fully-connected layer with a global average pooling layer, followed by a 1000-class fully-connected layer with softmax. To evaluate the impact of the network size, we explore the use of different dimensions, $[128, 256, 512]$, for the final convolutional features. We trained the resulting AlexNets and corresponding ExpandNets using the short-term regime of our MobileNets experiments in Section 3.

Table 9: **Top-1 accuracy** (%) **of AlexNet vs ExpandNets with** $r = 4$ **on the ILSVRC2012 validation set for different number of channels in the last convolutional layer.** Note that, while our expansion strategy always helps, its benefits decrease as the original model grows.

| # Channels | 128 | 256 (Original) | 512 |
|---|---|---|---|
| Baseline | 46.72 | 54.08 | 58.35 |
| ExpandNet-CK | **49.66** | **55.46** | **58.75** |

As shown in Table 9, while our approach outperforms the original AlexNet for all feature dimensions, the benefits decrease as the feature dimension increases. This indicates that our approach is

Table 10: Generalization ability (top-1 error (%)) on Corrupted CIFAR-10 (kernel size: 3, 7).

| Dataset | Model | Kernel size $k$ | | | | | |
| | | 3 | | | 7 | | |
| | | Best Test | Last Test | Train | Best Test | Last Test | Train |
| --- | --- | --- | --- | --- | --- | --- | --- |
| 20% | SmallNet | $22.20 \pm 0.43$ | $22.33 \pm 0.40$ | $34.85 \pm 0.18$ | $21.64 \pm 0.36$ | $21.98 \pm 0.42$ | $30.42 \pm 0.32$ |
| | ExpandNet-FC | $22.40 \pm 0.29$ | $22.61 \pm 0.27$ | $35.12 \pm 0.07$ | $21.92 \pm 0.23$ | $22.35 \pm 0.43$ | $30.39 \pm 0.19$ |
| | ExpandNet-CL | $\mathbf{21.55 \pm 0.27}$ | $\mathbf{21.71 \pm 0.30}$ | $34.89 \pm 0.26$ | $21.25 \pm 0.41$ | $21.54 \pm 0.40$ | $30.36 \pm 0.24$ |
| | ExpandNet-CK | $N/A$ | $N/A$ | $N/A$ | $\mathbf{19.11 \pm 0.33}$ | $\mathbf{19.30 \pm 0.35}$ | $\mathbf{31.14 \pm 0.11}$ |
| 50% | SmallNet | $25.74 \pm 0.25$ | $25.94 \pm 0.15$ | $56.48 \pm 0.25$ | $26.99 \pm 0.69$ | $27.87 \pm 0.71$ | $53.27 \pm 0.21$ |
| | ExpandNet-FC | $25.54 \pm 0.47$ | $25.80 \pm 0.41$ | $56.37 \pm 0.15$ | $26.86 \pm 0.46$ | $28.23 \pm 0.61$ | $53.14 \pm 0.20$ |
| | ExpandNet-CL | $\mathbf{25.48 \pm 0.35}$ | $\mathbf{25.66 \pm 0.43}$ | $56.41 \pm 0.33$ | $26.05 \pm 0.31$ | $26.99 \pm 0.15$ | $53.21 \pm 0.16$ |
| | ExpandNet-CK | $N/A$ | $N/A$ | $N/A$ | $\mathbf{22.43 \pm 0.47}$ | $\mathbf{22.61 \pm 0.49}$ | $\mathbf{53.74 \pm 0.16}$ |
| 80% | SmallNet | $37.49 \pm 0.62$ | $37.87 \pm 0.63$ | $77.46 \pm 0.16$ | $39.08 \pm 0.41$ | $43.33 \pm 0.77$ | $74.69 \pm 0.26$ |
| | ExpandNet-FC | $37.26 \pm 0.16$ | $37.63 \pm 0.14$ | $77.54 \pm 0.07$ | $40.51 \pm 0.39$ | $44.82 \pm 0.62$ | $75.38 \pm 0.23$ |
| | ExpandNet-CL | $\mathbf{35.86 \pm 0.43}$ | $\mathbf{36.05 \pm 0.44}$ | $\mathbf{77.56 \pm 0.11}$ | $39.40 \pm 0.93$ | $42.77 \pm 0.96$ | $75.24 \pm 0.22$ |
| | ExpandNet-CK | $N/A$ | $N/A$ | $N/A$ | $\mathbf{32.62 \pm 0.28}$ | $\mathbf{33.86 \pm 0.37}$ | $\mathbf{75.65 \pm 0.16}$ |

Table 11: Generalization ability (top-1 error (%)) on Corrupted CIFAR-100 (kernel size: 3, 5).

| Dataset | Model | Kernel size $k$ | | | | | |
| | | 3 | | | 5 | | |
| | | Best Test | Last Test | Train | Best Test | Last Test | Train |
| --- | --- | --- | --- | --- | --- | --- | --- |
| 20% | SmallNet | $55.30 \pm 0.42$ | $55.48 \pm 0.41$ | $62.20 \pm 0.31$ | $53.95 \pm 0.33$ | $54.16 \pm 0.34$ | $58.53 \pm 0.30$ |
| | ExpandNet-FC | $56.15 \pm 0.22$ | $56.35 \pm 0.23$ | $62.60 \pm 0.19$ | $54.84 \pm 0.71$ | $55.05 \pm 0.76$ | $59.47 \pm 0.32$ |
| | ExpandNet-CL | $\mathbf{54.85 \pm 0.27}$ | $\mathbf{55.04 \pm 0.33}$ | $61.62 \pm 0.43$ | $53.50 \pm 0.35$ | $53.71 \pm 0.38$ | $58.09 \pm 0.31$ |
| | ExpandNet-CK | $N/A$ | $N/A$ | $N/A$ | $\mathbf{51.98 \pm 0.28}$ | $\mathbf{52.10 \pm 0.26}$ | $57.67 \pm 0.63$ |
| 50% | SmallNet | $62.54 \pm 0.74$ | $62.71 \pm 0.75$ | $78.81 \pm 0.39$ | $61.84 \pm 0.29$ | $62.16 \pm 0.21$ | $76.78 \pm 0.28$ |
| | ExpandNet-FC | $63.65 \pm 0.50$ | $63.88 \pm 0.47$ | $79.40 \pm 0.18$ | $62.99 \pm 0.69$ | $63.21 \pm 0.60$ | $77.85 \pm 0.31$ |
| | ExpandNet-CL | $\mathbf{61.95 \pm 0.61}$ | $\mathbf{62.11 \pm 0.59}$ | $78.78 \pm 0.48$ | $61.49 \pm 0.39$ | $61.70 \pm 0.43$ | $76.73 \pm 0.26$ |
| | ExpandNet-CK | $N/A$ | $N/A$ | $N/A$ | $\mathbf{58.96 \pm 0.32}$ | $\mathbf{59.14 \pm 0.41}$ | $76.24 \pm 0.30$ |
| 80% | SmallNet | $78.35 \pm 0.83$ | $78.52 \pm 0.86$ | $93.78 \pm 0.18$ | $78.59 \pm 0.27$ | $78.81 \pm 0.35$ | $93.10 \pm 0.12$ |
| | ExpandNet-FC | $80.36 \pm 0.55$ | $80.47 \pm 0.55$ | $94.38 \pm 0.12$ | $80.97 \pm 0.51$ | $81.15 \pm 0.53$ | $94.10 \pm 0.16$ |
| | ExpandNet-CL | $79.44 \pm 0.72$ | $79.66 \pm 0.75$ | $94.02 \pm 0.16$ | $79.87 \pm 0.29$ | $80.04 \pm 0.29$ | $93.59 \pm 0.20$ |
| | ExpandNet-CK | $N/A$ | $N/A$ | $N/A$ | $\mathbf{77.22 \pm 0.47}$ | $\mathbf{77.38 \pm 0.41}$ | $93.15 \pm 0.25$ |

Table 12: Generalization ability (top-1 error (%)) on Corrupted CIFAR-100 (kernel size: 7, 9).

| Dataset | Model | Kernel size $k$ | | | | | |
| | | 7 | | | 9 | | |
| | | Best Test | Last Test | Train | Best Test | Last Test | Train |
| --- | --- | --- | --- | --- | --- | --- | --- |
| 20% | SmallNet | $55.36 \pm 0.44$ | $55.66 \pm 0.43$ | $56.33 \pm 0.51$ | $56.59 \pm 0.72$ | $57.26 \pm 0.64$ | $55.16 \pm 0.32$ |
| | ExpandNet-FC | $56.31 \pm 0.78$ | $56.58 \pm 0.77$ | $57.93 \pm 0.29$ | $57.82 \pm 0.23$ | $58.07 \pm 0.22$ | $57.09 \pm 0.52$ |
| | ExpandNet-CL | $54.87 \pm 0.47$ | $55.22 \pm 0.55$ | $55.52 \pm 0.49$ | $56.05 \pm 0.75$ | $56.51 \pm 0.76$ | $54.99 \pm 0.48$ |
| | ExpandNet-CK | $\mathbf{51.24 \pm 0.60}$ | $\mathbf{51.40 \pm 0.66}$ | $\mathbf{56.40 \pm 0.21}$ | $\mathbf{52.36 \pm 0.54}$ | $\mathbf{52.55 \pm 0.47}$ | $57.76 \pm 0.50$ |
| 50% | SmallNet | $63.76 \pm 0.59$ | $64.08 \pm 0.58$ | $75.45 \pm 0.23$ | $64.83 \pm 0.41$ | $65.63 \pm 0.40$ | $75.21 \pm 0.31$ |
| | ExpandNet-FC | $64.54 \pm 0.72$ | $64.91 \pm 0.55$ | $76.75 \pm 0.39$ | $66.11 \pm 0.45$ | $66.73 \pm 0.41$ | $76.44 \pm 0.40$ |
| | ExpandNet-CL | $63.36 \pm 0.49$ | $63.73 \pm 0.54$ | $75.25 \pm 0.45$ | $64.36 \pm 0.54$ | $65.25 \pm 0.37$ | $74.84 \pm 0.28$ |
| | ExpandNet-CK | $\mathbf{58.74 \pm 0.25}$ | $\mathbf{58.98 \pm 0.20}$ | $75.24 \pm 0.24$ | $\mathbf{60.42 \pm 0.86}$ | $\mathbf{60.65 \pm 0.86}$ | $\mathbf{76.73 \pm 0.40}$ |
| 80% | SmallNet | $79.73 \pm 0.47$ | $79.95 \pm 0.36$ | $92.58 \pm 0.20$ | $81.02 \pm 0.92$ | $81.70 \pm 0.97$ | $92.54 \pm 0.26$ |
| | ExpandNet-FC | $82.97 \pm 0.83$ | $83.20 \pm 0.83$ | $94.13 \pm 0.34$ | $83.42 \pm 0.72$ | $83.82 \pm 0.71$ | $93.94 \pm 0.35$ |
| | ExpandNet-CL | $80.79 \pm 0.54$ | $81.09 \pm 0.62$ | $93.22 \pm 0.30$ | $81.02 \pm 0.44$ | $81.58 \pm 0.46$ | $93.25 \pm 0.56$ |
| | ExpandNet-CK | $\mathbf{78.51 \pm 0.41}$ | $\mathbf{78.64 \pm 0.36}$ | $\mathbf{93.24 \pm 0.15}$ | $\mathbf{80.15 \pm 0.50}$ | $\mathbf{80.32 \pm 0.55}$ | $94.04 \pm 0.21$ |

better suited for truly compact networks, and developing similar strategies for deeper ones will be the focus of our future research.

## A.5 Generalization ability on Corrupted CIFAR-10 and CIFAR-100

Our experiments on corrupted datasets implies better generalization. Here, we provide more experimental results on Corrupted CIFAR-10 (in Table 10) and CIFAR-100 (in Table 11 and 12) by using different networks with kernel sizes of 3, 5, 7, 9, respectively. Our method consistently improves the generalization error gap across all kernel sizes and corruption rates (20%, 50%, 80%) and yields from around 1pp to over 6pp error drop in testing.

## B Complexity analysis

Here, we compare the network complexity of our method and the original networks in terms of number of parameters, number of FLOPs and GPU inference speed.

In Table 13, we provide numbers for both training and testing. During training, because our approach introduces more parameters, inference is 2 to 5 times slower than in the original network for an expansion rate of 4. However, since our ExpandNets can be compressed back to the original network,

Table 13: **ExpandNet complexity analysis on CIFAR-10, ImageNet, PASCAL VOC and Cityscapes.** Note that, within each task, the metrics are the same for all networks, since we can compress our ExpandNets back to the small network.

| Model | # Params(M) | | # FLOPs | | GPU Speed (imgs/sec) | |
|---|---|---|---|---|---|---|
| | Train | Test | Train | Test | Train | Test |
| SmallNet ($7 \times 7$) | 0.07 | | 4.49M | | 147822.51 | |
| ExpandNet-CL | 0.55 | | 57.49M | | 64651.81 | |
| ExpandNet-CL+FC | 2.11 | 0.07 | 59.04M | 4.49M | 61379.95 | 154850.52 |
| ExpandNet-CK | 0.19 | | 23.95M | | 75065.09 | |
| ExpandNet-CK+FC | 1.75 | | 25.5M | | 68679.89 | |
| MobileNet | 4.23 | 4.23 | 0.58G | 0.58G | 3797.21 | 3829.81 |
| ExpandNet-CL | 4.96 | | 1.76G | | 729.78 | |
| MobileNetV2 | 3.50 | 3.50 | 0.32G | 0.32G | 3417.20 | 3419.43 |
| ExpandNet-CL | 3.67 | | 1.34G | | 1009.25 | |
| ShuffleNetV2 $0.5\times$ | 1.37 | 1.37 | 0.04G | 0.04G | 5404.06 | 5434.58 |
| ExpandNet-CL | 1.41 | | 0.6G | | 4228.10 | |
| YOLO-LITE | 0.57 | 0.57 | 1.81G | 1.81G | 7.94 | 19.82 |
| ExpandNet-CL | 4.48 | | 28.59G | | 6.07 | |
| U-Net | 7.76 | 7.76 | 389.26G | 389.26G | 8.25 | 8.25 |
| ExpandNet-CL | 82.97 | | 2586.02G | | 2.98 | |

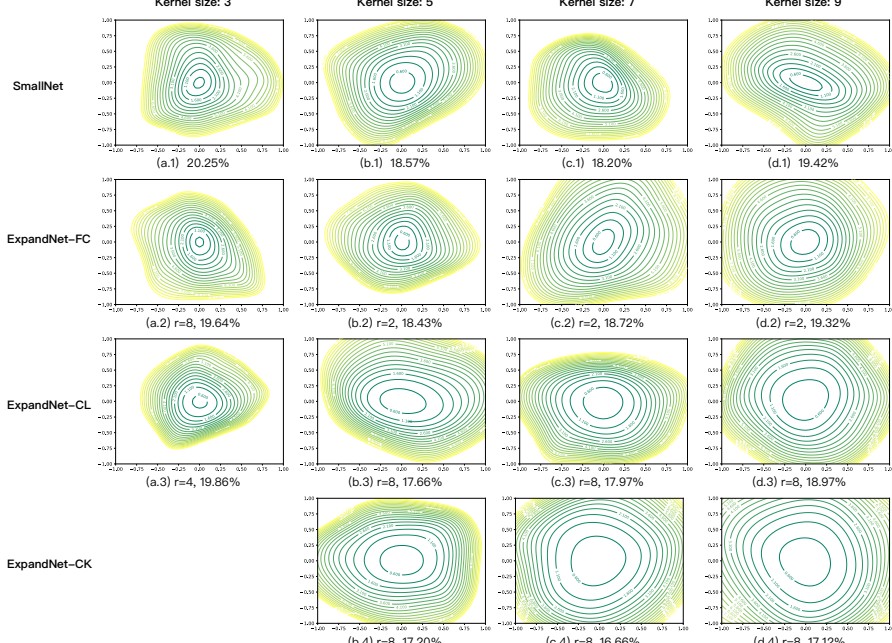

Figure 4: Loss landscape plots on CIFAR-10 (We report top-1 error (%)).

at test time, they have exactly the same number of parameters and FLOPS, and inference time, but our networks achieve higher accuracy.

## C   ADDITIONAL VISUALIZATIONS

Here, we provide additional visualizations for the training behavior and the loss landscapes of Section 3.4, corresponding to networks with kernel sizes of 3, 5, 7, 9, respectively.

We plot the loss landscapes of SmallNets and corresponding ExpandNets on CIFAR-10 in Figure 4, and analyze the training behavior on CIFAR-10 in Figure 5 and on CIFAR-100 in Figure 6, respectively. These plots further confirm that in almost all cases, our convolution expansion strategies (*CL* and *CK*) facilitate training (with lower gradient confusion and more 0-concentrated gradient cosine similarity density) and yield better generalization ability (with flatter minima).

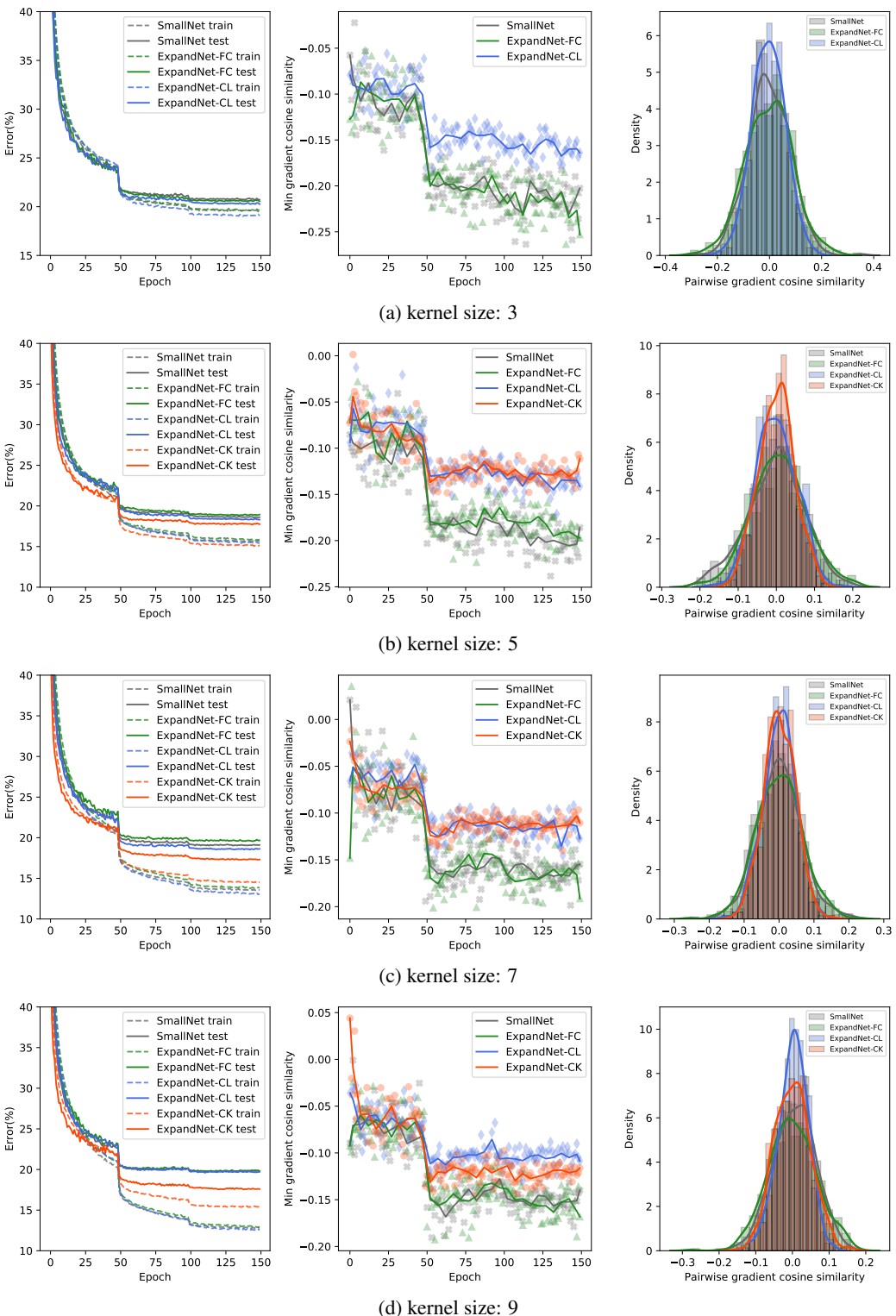

(a) kernel size: 3

(b) kernel size: 5

(c) kernel size: 7

(d) kernel size: 9

Figure 5: **Training behavior of networks on CIFAR-10.** *Left:* Training and test curves over 150 epochs. *Middle:* Minimum pairwise gradient cosine similarity at the end of each training epoch. *Right:* Kernel density estimation of pairwise gradient cosine similarity at the end of training (over 5 independent runs).

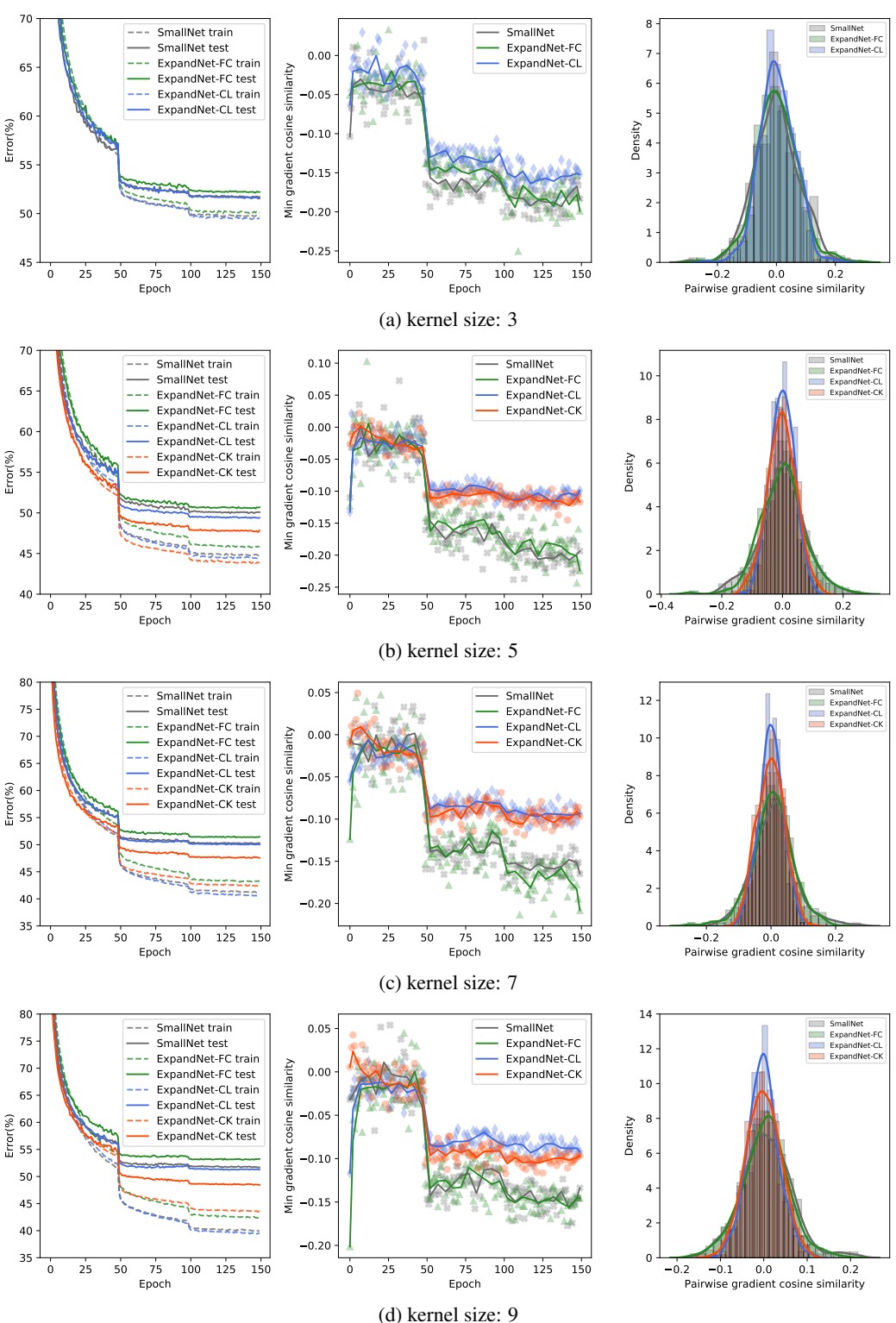

Figure 6: **Training behavior of networks on CIFAR-100.** *Left:* Training and test curves over 150 epochs. *Middle:* Minimum pairwise gradient cosine similarity at the end of each training epoch. *Right:* Kernel density estimation of pairwise gradient cosine similarity at the end of training (over 5 independent runs).

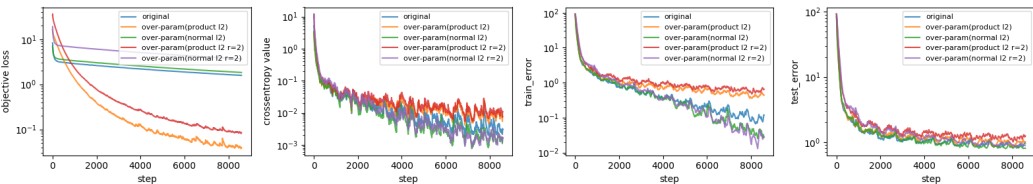

Figure 7: **Product $L^2$ vs Normal $L^2$** (best viewed in color). *Left:* Training curves of the overall loss function. *Middle Left:* Training curves of the cross-entropy loss. *Middle Right:* Curves of training errors. *Right:* Curves of test errors. (Note that the $y$-axis is in log scale.)

## D    DISCUSSION OF (ARORA ET AL., 2018)

In this section, we discuss in more detail the work of Arora et al. (2018) to further evidence the differences with our work. In (Arora et al., 2018), the authors only worked with linear models or linear layers (fully-connected layers). By contrast, we focus on practical, nonlinear, compact convolutional networks, and we propose 2 ways to expand convolutional layers, which has not been studied before. Our convolutional linear expansion strategies yield better solutions with higher accuracy, more zero-centered gradient cosine similarity during training and minima that generalize better.

In their paper, Arora et al. performed a sanity test on MNIST with a CNN, by only expanding the fully-connected layers. According to our experiments, with Expand-FC ONLY, getting better results than the compact network is difficult. Therefore, we used their code to reproduce their results, and found that, in their setting, the over-parameterized model yields higher test error. While analyzing the reasons for this, we observed that, in (Arora et al., 2018), given the input $(\boldsymbol{X}, \boldsymbol{Y})$, the loss function they used was defined as

$$Loss = CE(\boldsymbol{X}, \boldsymbol{Y}) + \lambda R, \tag{3}$$

$$R = \|\tilde{\boldsymbol{W}}_{fc}\|_2^2 \tag{4}$$

$$= \sum_i \|\tilde{\boldsymbol{W}}_{fc_i}\|_2^2 \,, \tag{5}$$

where $\tilde{\boldsymbol{W}}_{fc_i} = \prod_j \boldsymbol{W}_{fc_{i,j}}$ represents the product of the parameter matrices of the $i$-th over-parameterized fully-connected layer. *CE* denotes the cross-entropy loss and $\| \cdot \|_2$ is the $L^2$ norm. By contrast, the typical $L^2$ norm regularization used in deep learning is

$$R = \|\boldsymbol{W}_{fc}\|_2^2 \tag{6}$$

$$= \sum_i \sum_j \|\boldsymbol{W}_{fc_{i,j}}\|_2^2 \,, \tag{7}$$

which sums the square loss of the individual parameter matrices.

The $L^2$ norm regularizer used by Arora et al. (2018) imposes weaker constraints on the individual parameter matrices, and we observed their over-parameterized model to converge to a worse minimum during training and lead to worse test performance.

To evidence this, as shown in Figure 7, we compare the original model with an over-parameterized one as in (Arora et al., 2018), and with an over-parameterized network with normal $L^2$ regularization (corresponding to our ExpandNet-FC). Even though the overall loss of Arora et al. (2018)'s over-parameterized model decreases faster than that of the baseline, the cross-entropy loss, the training error and the test error do not show the same trend. The test errors of the original model, Arora et al. (2018)'s over-parameterized model with product $L^2$ norm and our ExpandNet-FC with normal $L^2$ norm are 0.9%, 1.1% and 0.8%, respectively. Furthermore, we also compare Arora et al. (2018)'s over-parameterized model and our ExpandNet-FC with expansion rate $r = 2$. We observed that Arora et al. (2018)'s over-parameterized model performs even worse with a larger expansion rate while our Expand-FC works well.

In short, the faster convergence suggested by Arora et al. (2018) seems to arise from the use of a different regularizer and does not lead to better test performance.

Table 14: Top-1 accuracy (%) of compact networks initialized with different ExpandNets on CIFAR-10, CIFAR-100 and ImageNet.

| Network | Initialization | CIFAR-10 | CIFAR-100 |
|---|---|---|---|
| SmallNet | Normal | $78.63 \pm 0.41$ | $46.63 \pm 0.27$ |
| | ExpandNet-FC | $79.09 \pm 0.56$ | $46.52 \pm 0.36$ |
| | ExpandNet-CL | $78.65 \pm 0.36$ | $46.65 \pm 0.47$ |
| | ExpandNet-CL+FC | $78.81 \pm 0.52$ | $46.43 \pm 0.72$ |
| | ExpandNet-CK | $78.84 \pm 0.30$ | $46.56 \pm 0.23$ |
| | ExpandNet-CK+FC | $79.27 \pm 0.29$ | $46.62 \pm 0.29$ |
| ExpandNet-CK+FC | Normal | $\mathbf{80.31 \pm 0.27}$ | $\mathbf{48.62 \pm 0.47}$ |

| | Initialization | ImageNet | |
|---|---|---|---|
| MobileNet | Normal | 66.48 | |
| MobileNet | ExpandNet-CL | 66.44 | |
| ExpandNet-CL | Normal | **69.40** | |
| MobileNetV2 | Normal | 63.75 | |
| MobileNetV2 | ExpandNet-CL | 63.07 | |
| ExpandNet-CL | Normal | **65.62** | |
| ShuffleNetV2 $0.5\times$ | Normal | 56.89 | |
| ShuffleNetV2 $0.5\times$ | ExpandNet-CL | 56.91 | |
| ExpandNet-CL | Normal | **57.38** | |

# E  ANALYSIS OF OUR EXPANDNETS

In this section, we conduct additional ablation study to reject some possible hypotheses and further evidence that the improvement of our approach is due to our expansion strategies.

**Hypothesis 1**: The improvement comes from initialization with the well-trained non-linear counterpart of the expanded network.

Based on the same experimental setting as in Table 1, we used the well-trained nonlinear counterparts to initialize the compact networks by computing the linear product of the expanded layers before training. The compact networks initialized using nonlinear ExpandNet-CL+FC achieve $77.02 \pm 0.35\%$ (vs $79.98 \pm 0.28\%$ using our initialization) on CIFAR-10 and $39.39 \pm 1.08\%$ (vs $47.98 \pm 0.48\%$ using our initialization) on CIFAR-100. Those initialized using nonlinear ExpandNet-CK+FC achieve $75.81 \pm 0.34\%$ (vs $80.81 \pm 0.27\%$ using our initialization) on CIFAR-10 and $34.88 \pm 1.41\%$ (vs $49.82 \pm 0.25\%$ using our initialization) on CIFAR-100.

Both experiments reveal that, while our initialization helps to improve our ExpandNet, using it to initialize the compact network is not effective.

**Hypothesis 2**: The improvement comes from initialization with the linear product of the expanded networks.

Following the experimental setting of Table 1 and Table 2, we initialize the compact networks with different expansion strategies on CIFAR-10, CIFAR-100 and ImageNet, respectively.

From reported results in Table 14 and Table 1, we can see that, on CIFAR-10, compact networks initialized by Expand-FC and Expand-CL yield slightly better results than training the ExpandNets by normal initialization. However, the same trend does not occur on CIFAR-100 and ImageNet, where, with ExpandNet-CL initialization, MobileNet (90 epochs) gets 66.44% (vs 66.48% for the baseline), MobileNetV2 (90 epochs) gets 63.07% (vs 63.75% for the baseline) and ShuffleNet gets 56.91% (vs 56.89% for the baseline). Moreover, compact networks initialized by ExpandNet-CK always yield worse results than training ExpandNet-CKs from scratch.

Therefore, we believe that the benefits of our expansion strategies cannot be obtained by initialization using ExpandNets.

