# OpenReview forum: "ExpandNets: Linear Over-parameterization to Train Compact Convolutional Networks"
_ICLR.cc/2020/Conference — Reject_

### Official Review · AnonReviewer3 · 2019-10-18
**Official Blind Review #3**

**Rating:** 3

**Review:**

This paper is extremely interesting and quite surprising. In fact, the major claim is that using a cascade of linear layers instead of a single layer can lead to better performance in deep neural networks. As the title reports, expanding layers seems to be the key to obtain extremely interesting results. Moreover, the proposed approach is extremely simple and it is well explained in Section 2 with equations (1) and (2). This paper can have a tremendous impact in the research in deep networks if results are well explained.

However, in its present form, it is hard to understand why the claim is correct. In fact, the model presented in the paper has a major obscure point. Equation (1) and (2) are extremely clear. Without non-linear functions, equations (1) and (2) describe a classical matrix factorization like Principal Component Analysis. Now, if internal matrices have more dimensions of the rank of the original matrix, the product of the internal matrices is exactly the original matrix. Whereas, if internal matrices have a number of dimensions lower than the rank of the original matrix, these matrices act as filters on features or feature combination. Since the authors are using inner matrices with a number of dimensions higher than the number of dimensions of the original matrix, there is no approximation and, then, no selection of features or feature combinations. Hence, without non-linear functions, where is the added value of the method? How the proposed method can have better results.
There are some possibilities, which have not been explored:
1) the performance improvement derives from the approximation induced by the representation of float or double in the matrices. The approximation act as the non-linear layers among linear layers.
2) the real improvement seems to be given by the initialization which has been obtained by using the non-linear counterpart of the expansion; to investigate whether this is the case, the model should be compared with a compact model where the initialization is obtained by using the linear product of the non-linear counterpart of the expanded network. If this does not lead to the same improvement, there should be a value in the expansion.
3) the small improvement of the expanded network can be given by the different initialization. In fact, each composing matrix is initialized randomly. The product of a series of randomly initialized matrices can lead to a matrix that is initialized with a different distribution where, eventually, components are not i.i.d.. To show that this is not relevant, the authors should organize an experiment where the original matrix (in the small network) is initialized with the dot product of the composing matrices. The training should be done by using the small network. If results are significantly different, then the authors can reject the hypothesis.
If the authors can reject (1), (2) and (3), they should find a plausible explaination why performance improves in their experiments.

**Experience Assessment:**

I have published one or two papers in this area.

**Review Assessment: Checking Correctness Of Derivations And Theory:**

I assessed the sensibility of the derivations and theory.

**Review Assessment: Checking Correctness Of Experiments:**

I assessed the sensibility of the experiments.

**Review Assessment: Thoroughness In Paper Reading:**

I read the paper at least twice and used my best judgement in assessing the paper.

---

> ### Author Response · Authors · 2019-11-14
> **Response to Review #3 (Part 1/2)**
>
>
> Thank you for your insightful and valuable review. We address your comments below and have modified our paper accordingly. We would appreciate any further feedback.
>
> Q: Without non-linearity, what is the added value and how we have better results?
>
> Over-parameterization has been shown both theoretically and empirically to facilitate neural network training. We believe this to be exactly our contribution: discovering a simple but effective method that takes advantage of over-parameterization during training by expanding layers, while this over-parameterization is not necessary at inference so that we can compress the better trained expanded networks back to original ones without losing any performance.  Note that it is not the same as naively adding new layers to the network with nonlinearity, which would give no direct way to do a lossless compression afterwards. This is a major contribution since no one has studied this aspect before.
>
> As suggested, we conducted additional experiments and analysis to reject the proposed hypotheses and further evidence that the improvement is due to our expansion strategies.
>
> 1) Q:  The improvements are possibly from implicit nonlinear operations between expanded layers.
>
> We conducted experiments to compare the representation ability of float (32 bit) and double (64 bit) precision and to simulate the nonlinearity arising from truncation between expanded layers. Based on the experimental setting of Table 1, but with convolutional kernel size = 5, we trained a small network with float and double as baseline, and then trained an ExpandNet-CK with r=4 in float. In addition, we truncated the small network in double precision to float during training, to simulate the nonlinearity arising when multiplying 2 floats into 1 float. To be precise, we used double  to compute (W*x + b), and then converted the result to float to truncate the output feature maps.
>
> --------------------------------------------------------------------------------------------
>    Network                              |     CIFAR-10      |    CIFAR-100
> --------------------------------------------------------------------------------------------
>    SmallNet(float)			|   78.94 ± 0.40   |   47.33 ± 0.46
>    SmallNet(double)		|   78.73 ± 0.48   |   46.28 ± 0.50
>    SmallNet(double) with
>    truncated non-linearity	|   78.98 ± 0.13   |   46.21 ± 0.94
>    ExpandNet-CK (float)    	|   79.90 ± 0.38   |   48.26 ± 0.14
> --------------------------------------------------------------------------------------------
>
> As shown by the results reported in the table above, the networks with double precision do not outperform those with float precision. Furthermore, the nonlinearity obtained by truncation does not help during training. Our ExpandNet-CK consistently outperforms SmallNets with float or double or truncation. At test time, the outputs from an (uncompressed) ExpandNet and one that was compressed back to the SmallNet architecture are equal, up to precision error, which indicates that truncation does not affect the test performance.
>
> Thus we can conclude that our improvement is not due to higher numerical precision or the implicit truncation nonlinearity arising from taking the products of several floats in the expanded linear layers.
>
> 2) Q: Initialize the compact networks with the linear product of the well-trained non-linear counterpart of the expanded network.
>
> This is an interesting suggestion. Based on the same experimental setting as in Table 1, we used the well-trained nonlinear counterparts to initialize the compact networks by computing the linear product of the expanded layers before training as suggested. The compact networks initialized using nonlinear ExpandNet-CL+FC achieve 77.02 ± 0.35% (vs 79.98 ± 0.28% using our initialization) on CIFAR-10 and 39.39 ± 1.08% (vs 47.98 ± 0.48% using our initialization) on CIFAR-100. Those initialized using nonlinear ExpandNet-CK+FC achieve 75.81 ± 0.34% (vs 80.81 ± 0.27% using our initialization) on CIFAR-10 and 34.88 ± 1.41% (vs 49.82 ± 0.25% using our initialization) on CIFAR-100.
>
> Both experiments reveal that, while our initialization helps to improve our ExpandNet, using it to initialize the compact network is not effective.

---

> ### Author Response · Authors · 2019-11-14
> **Response to Review #3 (Part 2/2)**
>
>
>
> 3) Q: Initialize the compact networks with the linear product of the expanded networks.
>
> We tried this before and found that it was not the reason behind the improvements. Following the experimental setting of Table 1 and Table 2, we initialized the compact networks with different expansion strategies on CIFAR-10, CIFAR-100 and ImageNet, respectively. The results are given in the table below.
>
> -------------------------------------------------------------------------------------------------------------------------
>    Network                            | Initialization       	           |       CIFAR-10        |     CIFAR-100
> -------------------------------------------------------------------------------------------------------------------------
>    SmallNet                            | Normal                           |     78.63 ± 0.41     |    46.63 ± 0.27
>    SmallNet                            | ExpandNet-FC               |     79.09 ± 0.56     |    46.52 ± 0.36
>    SmallNet                            | ExpandNet-CL               |     78.65 ± 0.36     |    46.65 ± 0.47
>    SmallNet                            | ExpandNet-CL+FC         |     78.81 ± 0.52     |    46.43 ± 0.72
>    SmallNet                            | ExpandNet-CK               |     78.84 ± 0.30     |    46.56 ± 0.23
>    SmallNet                            | ExpandNet-CK+FC        |     79.27 ± 0.29     |    46.62 ± 0.29
>    ExpandNet-CK+FC(ours) | Normal                           |     80.31 ± 0.27     |    48.62 ± 0.47
> -------------------------------------------------------------------------------------------------------------------------
>
> From these results and results in Table 1, we can see that, on CIFAR-10, compact networks initialized by Expand-FC and Expand-CL yield slightly better results than training the ExpandNets by normal initialization. However, the same trend does not occur on CIFAR-100 and ImageNet, where, with ExpandNet-CL initialization, MobileNet (90 epochs) gets 66.44% (vs 66.48% for the baseline and 69.40% for the ExpandNet-CL), MobileNetV2 (90 epochs) gets 63.07% (vs 63.75% for the baseline and 65.62% for the ExpandNet-CL) and ShuffleNet gets 56.91% (vs 56.89% for the baseline and 57.38% for the ExpandNet-CL). Moreover, compact networks initialized by ExpandNet-CK always yield worse results than training ExpandNet-CKs from scratch.
>
> Therefore, we believe that the benefits of our expansion strategies cannot be obtained by initialization using ExpandNets.
>
> 2) and 3)  are great points that we have added them to our paper in Appendix E . Thanks again for these suggestions.

---

### Official Review · AnonReviewer2 · 2019-10-28
**Official Blind Review #2**

**Rating:** 6

**Review:**

This paper proposes linear over-parameterization methods to improve training of small neural network models. The idea is simple -- each linear transformation in a network is overparameterized by a series of linear transformation which is algebraically equivalent to the original linear transformation. Number of experiments are conducted to show the effectiveness of the approach.

The proposed method is a simple application of over-parameterization to improve neural network model training. The motivation is clear and the proposed method is clearly presented. The paper is easy to understand and follow. Great analyses on the training behavior and generalization ability are conducted. Given the simplicity of the method, this could be a standard way of training small neural network model if the effectiveness of the method is observed more widely.

These are some concerns on the paper:

1) The effectiveness of the approach is not necessarily significant in all experiments. For example, in Table 1, ExpandNet-FC and ExpandNet-CL were not effective. The same trend is observed in Table 2 for 400 epochs. Given that only ExpandNet-CK improves the performance, we could conclude that some intrinsic property of CK is important than over-parameterization. The good results for 90 epochs in Table 2 may mean linear over-parameterization yields faster convergence as suggested by Arora et al. 2018.

2) The comparisons are not extensive. For example, we do not see Init for all models and "w/ KD" for ShuffleNetV2 in Table 2. Table 2 has "N/A". Table 3 and 4 do not have results with "w/ KD".  Knowledge transfer methods should be the baseline of the paper.

Minor comments:

It will be interesting to see the results of the models used for Init.

It might be interesting to conduct experiments with a big model and see if we do not have any gains.

**Experience Assessment:**

I have read many papers in this area.

**Review Assessment: Checking Correctness Of Derivations And Theory:**

I assessed the sensibility of the derivations and theory.

**Review Assessment: Checking Correctness Of Experiments:**

I carefully checked the experiments.

**Review Assessment: Thoroughness In Paper Reading:**

I read the paper at least twice and used my best judgement in assessing the paper.

---

> ### Author Response · Authors · 2019-11-14
> **Response to Review #2 (Part 1/2)**
>
>
> Thank you for your comments. We address your concerns below.
>
> 1) Effectiveness of our approach.
>
> Expanding FC, CL or CK are different design choices of our approach, and we show them all in the paper for completeness. We demonstrate their effectiveness on 3 tasks (image classification, object detection, and semantic segmentation), using 5 different datasets  (ImageNet, PASCAL VOC, Cityscapes, CIFAR-10, CIFAR-100), and with 7 different compact network architectures (SmallNet3x3, SmallNet7x7, MobileNet, MobileNetV2, ShuffleNet, YOLO-LITE, U-Net). It is clear that our ExpaneNets are effective in all experimental settings. Although one can always run more experiments, this already constitutes strong evidence that our method works.
>
> We nevertheless discuss your individual comments in more detail below.
>
>     1.1) Q: Expand-FC and Expand-CL are not significant in Table 1, and Table 2 for 400 epochs.
>
> To be exact, ExpandNet-FC is the only configuration that does not achieve better results. However, it is merely one of our configurations. As shown in Appendix A.3 with multiple expansion rates and networks, ExpandNet-CL consistently outperforms the baselines. In addition, expanding FC and CL together consistently improves the training of small networks, as shown in Table 1. We also demonstrate the effectiveness of ExpandNet-CL on many large-scale image recognition tasks (classification using ImageNet; detection using Pascal VOC, segmentation using Cityscapes). This constitutes strong evidence that our approach is effective.
>
> Furthermore, as shown in Figure 2 and 3, ExpandNet-CL produces flatter minima and more zero-centered gradient cosine similarity, which indicates that ExpandNet-CL has a better training behavior and can reach solutions that generalize better. This is an important property of our convolutional expansion.
>
>
>     1.2) Q: Some intrinsic property of Expand-CK is more important than over-parameterization.
>
> We conducted an additional ablation study for ExpandNet-CK to show that both ExpandNet-CK and over-parameterization are important.
>
> In our method, the expansion rate (r) controls the number of parameters of ExpandNet-CK. Following the same experimental setting as for Table 1, where the baseline SmallNet achieves a top-1 accuracy of 78.63% on CIFAR-10 and 46.63% on CIFAR-100, we set the expansion rate in [0.25, 0.5, 0.75, 1.0, 2.0, 4.0], and report the corresponding results in the following table. For r < 1, the performance of ExpandNet-CK drops from 78.70% to 72.32% on CIFAR-10 and from 46.41% to 39.32% on CIFAR-100 as the number of parameters decreases. For r > 1, ExpandNet-CK yields consistently higher accuracy. Interestingly, with r = 1, ExpandNet-CK still yields better performance (79.22% vs 78.63% and 47.25% vs 46.63%) with fewer parameters (54.77K vs 66.19K and 60.62K vs 72.04K).
>
> --------------------------------------------------------------------------------------------
>    expansion rate |    #params (K)  |     CIFAR-10      |    CIFAR-100
> --------------------------------------------------------------------------------------------
>    0.25                    |   37.91 / 43.76   |   72.32 ± 0.62   |   39.23 ± 0.84
>    0.50                    |   42.81 / 48.66   |   76.77 ± 0.36   |   43.68 ± 0.51
>    0.75                    |   48.43 / 54.28   |   78.70 ± 0.42   |   46.41 ± 0.52
>    1.00                    |   54.77 / 60.62   |   79.22 ± 0.52   |   47.25 ± 0.40
>    SmallNet           |   66.19 / 72.04   |   78.63 ± 0.41   |   46.63 ± 0.27
>    2.00                    |   87.32 / 93.17   |   79.97 ± 0.18   |   48.13 ± 0.42
>    4.00                    |  186.98 / 192.8  |   80.27 ± 0.24   |   48.55 ± 0.51
> --------------------------------------------------------------------------------------------
> (#params(K) denotes the number of parameters: CIFAR-10 / CIFAR-100)
>
> Altogether, these results indicate that, while ExpandNet-CK in itself helps to improve the results, combining it with over-parameterization further boosts the performance.
>
> We also conducted an ablation study using different expansion rates and networks in Appendix A.3, Table 8, to investigate the importance of over-parameterization.
>
> Note that all the parameter numbers refer to the training parameters; at test time, our ExpandNets are compressed back, without any loss, so as to have the same number of parameters as the baseline compact network.

---

> ### Author Response · Authors · 2019-11-14
> **Response to Review #2 (Part 2/2)**
>
>
>     1. 3) Q: The proposed method is a simple application of over-parameterization. The good results might yield from faster convergence of linear over-parameterization as suggested by Arora et.al. 2018.
>
> Arora et al. 2018 only worked with linear models or linear layers. By contrast, we focus on practical, nonlinear, compact convolutional networks, and we propose to expand convolutional layers, which has not been studied before. Exploring how to expand convolutional layers is one of our contributions.
>
> In their paper, Arora et al. performed a sanity test on MNIST with a CNN, by only expanding the fully-connected layers. According to our experiments, with Expand-FC ONLY, getting better results than the compact network is difficult.  In Appendix D, we perform a more thorough evaluation of the behavior observed by Arora et al. In short, the faster convergence they observed seems to be due to their use of a different regularizer, acting on the product of the parameter matrices of the expanded layers, rather than on the individual parameters. This, in turn, makes their model yield worse test error than the compact network, whereas our ExpandNets, which rely on standard regularization, achieve better results. See Appendix D of the revised paper for the detailed discussion.
>
> 2) Missing results of 400 epochs and KD on ShuffleNetV2, Table 3 and Table 4.
>
> Here are the results of KD on ShuffleNet: ShuffleNet (w/KD) achieves 57.59% and ExpandNet-CL (w/KD) achieves 57.68% [ShuffleNet yields 56.89% and ExpandNet-CL 57.38%]. We are waiting for the results of 400 epochs. We will include these results in our paper when we get all of them.
>
> We tend to disagree that knowledge transfer methods should be our main baselines. Our approach is complementary to knowledge transfer, and it can also be used on its own in the absence of teacher networks. In any event, Table 1 and 2 already indicate that, in most cases, baseline < baseline+KD < ExpandNet < ExpandNet+KD in terms of accuracy. The ShuffleNet results above confirm that the performance of our ExpandNets can be further boosted with the help of a teacher network.
>
> Note that using KD or knowledge transfer with YOLO and U-Net is not straightforward and has received very little attention so far. Doing so goes beyond the scope of this work.
>
> 3) Initialize all models and apply the methods widely to big models.
>
> In our experiments, we found that, on some datasets, the ExpandNets’ nonlinear counterparts do not outperform the original models. Using these as initialization does not provide a good starting point. In other words, nonlinearity does not always help in deep networks and our initialization works much better when the baseline networks are quite small.
>
> We did conduct some experiments on deeper and wider networks, but the improvements are not significant. As shown in Appendix A.4, Table 9, where we investigate the use of our Expand-CK on AlexNet with different number of channels, we found that the benefits decrease as the compact model size increases. This, we believe, further evidences that the benefits of our approach are due to over-parameterization.

---

### Author Response · Authors · 2019-11-14
**Summary of Changes in New Version**


We thank the reviewers for their valuable comments. We have revised our paper in the following way:

1. As suggested by reviewer 2, we have added knowledge distillation for ShuffleNet in Table 2.

2. As suggested by reviewer 2, we have added Appendix D to discuss the work of Arora et al. (2018) in detail and further highlight the differences with our work.

3. As suggested by reviewer 3, we have added the analysis of our expansion strategies to Appendix E.

---

### Decision · Program_Chairs · 2019-12-19

**Decision:**

Reject

**Comment:**

The paper develops linear over-parameterization methods to improve training of small neural network models. This is compared to training from scratch and other knowledge distillation methods.

Reviewer 1 found the paper to be clear with good analysis, and raised concerns on generality and extensiveness of experimental work. Reviewer 2 raised concerns about the correctness of the approach and laid out several other possibilities. The authors conducted several other experiments and responded to all the feedback from the reviewers, although there was no final consensus on the scores.

The review process has made this a better paper and it is of interest to the community. The paper demonstrates all the features of a good paper, but due to a large number of strong papers, was not accepted at this time.